# Differing structures of galactoglucomannan in eudicots and non-eudicot angiosperms

**Konan Ishida**[1], **Yusuke Ohba**[2], **Yoshihisa Yoshimi**[1], **Louis F. L. Wilson**[1¤],
**Alberto Echevarría-Poza**[1], **Li Yu**[1], **Hiroaki Iwai**[3], **Paul Dupree**[1] *

**1** Department of Biochemistry, University of Cambridge, Hopkins Building, The Downing Site, Tennis Court
Road, Cambridge, United Kingdom, **2** Graduate School of Life and Environmental Science, University of
Tsukuba, Tsukuba, Ibaraki, Japan, **3** Institute of Life and Environmental Sciences, University of Tsukuba,
Tsukuba, Ibaraki, Japan

¤ Current address: Department of Molecular Physiology and Biophysics, University of Virginia,
Charlottesville, Virginia, United States of America
* pd101@cam.ac.uk

pone.0289581

UNITED STATES

**Data Availability Statement:** All relevant data are
within the manuscript and its Supporting
Information files. original gels and minimal data are

## Abstract

The structures of cell wall mannan hemicelluloses have changed during plant evolution.
Recently, a new structure called β-galactoglucomannan (β-GGM) was discovered in eudicot
plants. This galactoglucomannan has β-(1,2)-Gal-α-(1,6)-Gal disaccharide branches on
some mannosyl residues of the strictly alternating Glc-Man backbone. Studies in Arabidopsis revealed β-GGM is related in structure, biosynthesis and function to xyloglucan. However, when and how plants acquired β-GGM remains elusive. Here, we studied mannan
structures in many sister groups of eudicots. All glucomannan structures were distinct from
β-GGM. In addition, we searched for candidate mannan β-galactosyltransferases (MBGT)
in non-eudicot angiosperms. Candidate *At*MBGT1 orthologues from rice (*Os*GT47A-VII)
and Amborella (*Atr*GT47A-VII) did not show MBGT activity *in vivo*. However, the *At*MBGT1
orthologue from rice showed MUR3-like xyloglucan galactosyltransferase activity in complementation analysis using Arabidopsis. Further, reverse genetic analysis revealed that the
enzyme (*Os*GT47A-VII) contributes to proper root growth in rice. Together, gene duplication
and diversification of GT47A-VII in eudicot evolution may have been involved in the acquisition of mannan β-galactosyltransferase activity. Our results indicate that β-GGM is likely to
be a eudicot-specific mannan.

## Introduction

Mannans are polysaccharides used as a structural component in plant cell walls, as well as for
energy storage. Depending on plant lineage, developmental stage, and tissue, a variety of mannan structures has been reported such as homomannan, galactomannan, glucomannan, and
galactoglucomannan [1]. All of these have a backbone of β-(1,4)-linked residues. Mannose
(Man) is the major backbone component but some types of mannan have glucose (Glc) interspersed at different proportions (Exception: β-galactoglucomannan, discussed below, has a
repeating Man and Glc backbone). Mannan often has α-(1,6)-galactose (Gal) side chains. In

in the university of Cambridge archive https://doi.org/10.17863/CAM.101707.

**Funding:** This work was supported by the ERC Advanced Grant EVOCATE to P.D. funded by the United Kingdom Research and Innovation (UKRI) grant number EP/X027120/1 (www.ukri.org). The Japan Society for the Promotion of Science KAKENHI (Grant Nos. 23K05487 and 20K05950 www.jsps.go.jp) to H.I. and the 31st and 32nd Botanical Research Grant of ICHIMURA Foundation for New Technology (www.sgkz.or.jp) to H.I supported the work of H.I. and U.O. K.I. was supported by the Masayoshi-Son foundation scholarship (masason-foundation.org). A Broodbank Research Fellowship of the University of Cambridge (www.cam.ac.uk) no. PD16178 supported Y.Y. The University of Cambridge Herchel Smith scholarship supported A.E (www.cam.ac.uk). The funders had no role in study design, data collection and analysis, decision to publish, or preparation of the manuscript.

**Competing interests:** The authors have declared that no competing interests exist.

addition to the α-Gal decoration, the di-galactosylated side chain [β-(1,2)-Gal-α-(1,6)-Gal] has also been discovered in several eudicots [2–4]. These decorations may help to increase the solubility and interaction with cellulose [5, 6]. Mannan biosynthesis mutants show defects in seed mucilage extrusion [7, 8], embryogenesis and pollen tube growth [9], and vegetative growth in plants lacking xyloglucan [4]. However, we do not know how structural diversity in mannans are linked to the function. Therefore, revealing mannan structure-function relationships is important for a better understanding of the physiological roles of these polysaccharides.

Much has been learned about mechanisms of mannan biosynthesis using genetic studies in *Arabidopsis thaliana*. Arabidopsis studies revealed that at least three glycosyltransferase families are involved in mannan biosynthesis: CELLULOSE SYNTHASE-LIKE A (CSLA, GT2), MANNAN α-GALACTOSYLTRANSFERASE (MAGT, GT34), and MANNAN β-GALACTOSYLTRANSFERASE (MBGT, GT47A-VII) [1, 4, 10–14]. In addition, some types of mannan have an *O*-acetyl group on the backbone Man [15–17]. Mannan *O*-acetylation is catalysed by the member of TRICHOME BIREFRINGENCE-LIKE family, similar to xylan *O*-acetylation [18, 19]. Arabidopsis has nine CSLA genes, two of which (CSLA2 and CSLA9) synthesise most of the mannans in many tissues [4, 9]. Mannan synthesised by CSLA9 is called Acetylated GalactoGlucoMannan (AcGGM). This has a random order of Man and Glc backbone, and the Man residues are often acetylated [15, 18]. In Arabidopsis tissues, there are few or no α-Gal side chains on AcGGM. On the other hand, mannan synthesised by CSLA2 is called β-GalactoGlucoMannan (β-GGM). This has a strictly repeating backbone composed of [4)-Man-β-(1,4)-Glc-β-(1] units and has additionally a di-galactosylated side chain (Fig 1A). A recent study revealed that β-GGM is present in many eudicots, and showed that β-GGM has a partially overlapping function with another major hemicellulose, xyloglucan, in primary cell wall-rich tissues [4]. The structural and functional similarities are underpinned by the analogous biosynthesis enzymes [4]. In the case of xyloglucan, CELLULOSE SYNTHASE-LIKE C (CSLC, GT2), XYLOGLUCAN XYLOSYLTRANSFERASE (XXT, GT34), and one of XYLOGLUCAN GALACTOSYLTRANSFERASE (XLT, GT47A-III) or MURUS3 (MUR3, GT47A-VI) are required for one of the major side chain structure (Gal-β-1,2-Xyl-α-1,6-Glc) [4, 20]. Considering the widespread presence of β-GGM in eudicot [4], MBGT should also be conserved in GT47A. However, only a single MBGT (*At*MBGT1) gene has been identified from all plant species so far [4].

It is not clear whether β-GGM also exists outside of eudicots, nor whether the different glucomannans have similar functions. Angiosperms are divided into four groups: eudicots, monocots, magnoliids, and other orders (Amborellales, Nymphaeales, and Austrobaileyales: here we refer to them as "ANA grade"). Apart from eudicots, mannan structures from some monocot species have been reported. Banana (*Musa* sp, Zingiberales) pulp contains homomannan and glucomannan [21]. Palm (*Phoenix dactylifera*, Arecales) seeds are dominantly composed of homomannan [22]. In Arecales, galactomannan and pure mannan from coconuts (*Cocos nucifera*) kernel [23], and crystalised homomannan from ivory nuts (*Phytelephas macrocarpa*) have been characterised [24, 25]. As storage polysaccharides, konjac (*Amorphophallus konjac*, Alismatales) corm glucomannan, orchid (*Dendrobium officinale* and *Oncidium* cv. Gower Ramsey, Asparagales) stem glucomannan, aloe (*Aloe vera*, Asparagales) acetylated mannan, and wheat (*Triticum aestivum*, Poales) endosperm short homomannan were well studied [13, 26–29]. These indicate that at least homomannan, glucomannan, galactomannan, galactoglucomannan, and acetylated mannan are present in monocot species.

Here, we studied mannan structure from 21 species of monocot and ANA-grade plants to further characterise the distribution of β-GGM in angiosperms. After mannanase digestion, non-eudicot angiosperms showed different mannan fingerprints distinct from eudicots. Also, these sister groups did not have β-GGM-derived digestion products. Furthermore, we tested

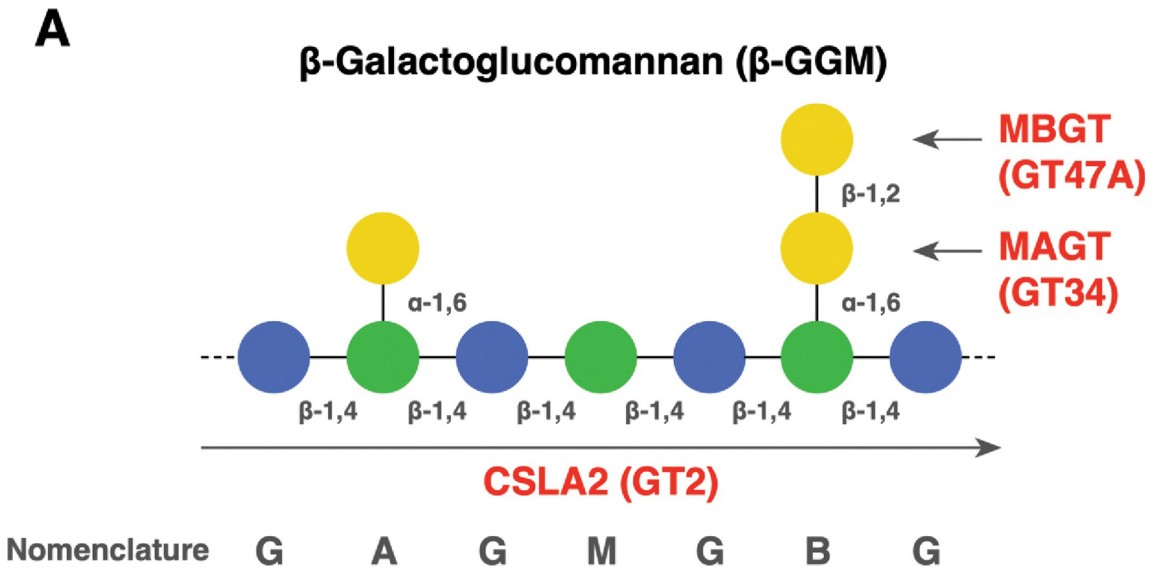

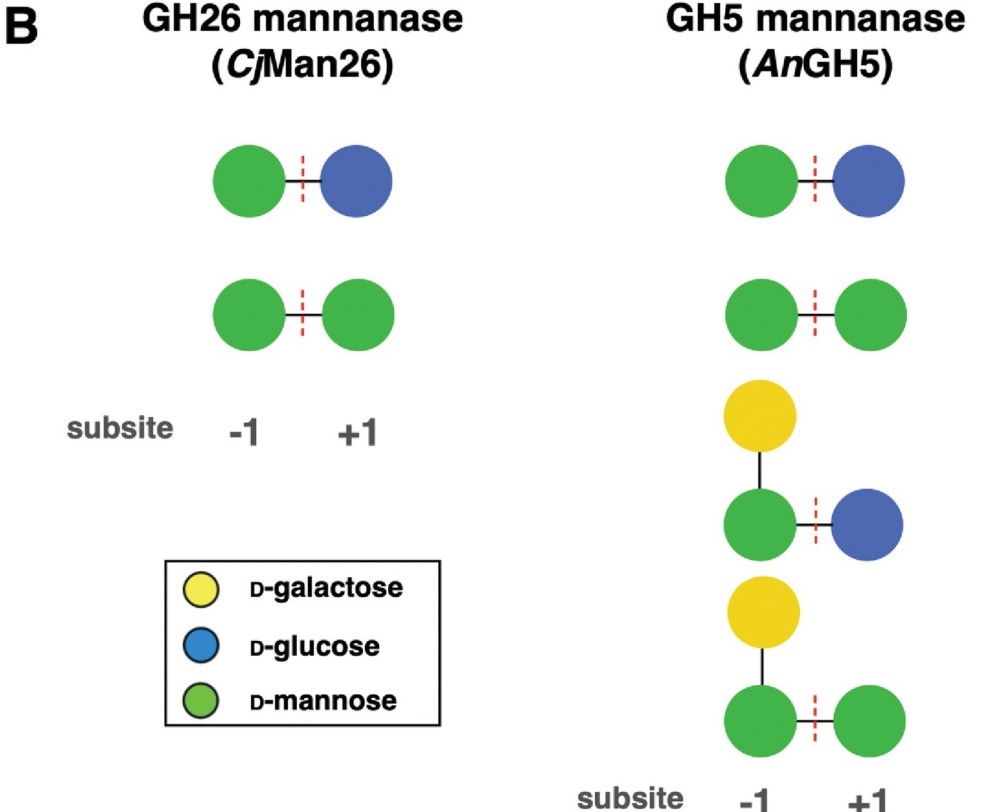

**Fig 1. Structure of β-GGM and substrate specificity of mannanases.** (A) β-GGM structure, the single-letter codes, and CAZy families of Arabidopsis biosynthesis enzymes [4]. (B) The GH26 mannanase (*Cj*Man26) and GH5 mannanase (*An*GH5) have different substrate specificity. Both require Man residue at subsite -1 and accommodate either Glc or Man at subsite +1. Only *An*GH5 allows the α-Gal side chain at -1 position.

the *in vivo* biochemical activity of the *At*MBGT1 orthologues from rice and Amborella. These orthologues did not exhibit MBGT activity. Therefore, these results suggest the possibility that β-GGM is a eudicot-specific mannan, and acquisition of MBGT activity was one of the key steps for β-GGM production among eudicots.

## Materials and methods

### Plant materials

21 species (39 samples) of monocot and ANA-grade samples (listed in Table 1) were collected at the University of Cambridge Botanic Garden in November 2020. *Arabidopsis thaliana mbgt1* mutant (*mbgt1-1*, SALK_065561) was studied in a previous report [4]. *Arabidopsis thaliana mur3* mutant (*mur3-1*, CS8566), *xlt2* mutant (GABI_552C10), and *mur3 xlt2* double mutant (*mur3-1 xlt2*) were studied in a previous report [30]. MicroTom variety of tomatoes (*Solanum lycopersicum*) was used in this study. Arabidopsis and tomato were grown at 21˚C under 16-hour light and eight-hour dark conditions. Rice was grown in a hydroponic medium under 16-hour light condition (30˚C,250 µmol m$^{-2}$ s$^{-1}$) and 8-hour dark conditions (28˚C, 0 µmol m$^{-2}$ s$^{-1}$). The composition of the hydroponic medium was 1 mM $KNO_3$, 0.5 mM $NH_4H_2PO_4$, 0.5 mM $MgSO_4$, 0.5 mM $CaCl_2$, 25 mM Fe-EDTA and micronutrients [31].

### Preparation of hemicellulose fraction

Soluble hemicelluloses were prepared as previously described [32]. Briefly, plant materials were soaked in 96% EtOH in 15 mL tube for three days. Then, the materials were cut into small pieces with scissors and homogenised by a ball mixer in EtOH. The ground samples were transferred to another 15 ml tube and added 10 ml 96% EtOH. The mixture was centrifuged at 4000 × g for 15 minutes. The supernatant was removed and added 10 ml non-polar solution (MeOH:$CHCl_3$ = 2v:3v). The mixture was shaken for one hour at 25˚C. Subsequently, the supernatant was removed by the centrifugation condition as mentioned above and the non-polar solution wash was repeated. The plant tissues were then washed in stages with 100%, 65%, 80% and 100% EtOH, and the alcohol insoluble residue (AIR) was completely dried. 100 mg AIR was de-pectinated in 50 mM ammonium oxalate for 1 hour. After the centrifugation at 20000 × g for five minutes, the supernatant was removed. To obtain the hemicellulose fraction, the sediment was homogenized in 4 M KOH and incubated at 25˚C for one hour. After the centrifugation at 20000 × g for five minutes, the supernatant was collected as hemicellulose fraction. This fraction was applied to a PD-10 column (GE Healthcare, IL, USA) to exchange the buffer for 50 mM ammonium acetate (pH 6.0) for the following enzyme reaction.

### Enzyme digestion

Crude hemicellulose mixture was digested with an excess amount of mannanase [*Cellvibrio japonicus* Man26A (*Cj*Man26) or *Aspergillus nidulans* GH5 (*An*GH5)] at 37˚C overnight (1 µl enzyme was used for the 1 mg AIR equivalent PD-10 eluate) (substrate specificities of two mannanases are shown in Fig 1B). Before further digestion was carried out, the reaction mixture was heated at 100˚C for 20 minutes to de-activate the mannanase activity. Then, other enzymes such as β-galactosidase (from *Aspergillus niger*, GH35), α-galactosidase (from *Cellvibrio mixtus*, GH27), β-glucosidase (from *Aspergillus niger*, GH3), or β-mannosidase (from *Cellvibrio mixtus*, GH5) was added to the same excess volume and incubated at 37˚C for four hours per reaction. The reaction mixture was heated (100˚C for 20 minutes) every time before adding the next enzyme.

**Table 1. List of plant samples studied in this work.**

| No. | Group | Orders | Species | Tissues |
|---|---|---|---|---|
| 1 | Monocots | Zingiberales | *Zingiber spectabile* | Leaf |
| 2 | | | | Stem |
| 3 | | | *Musa basjoo* | Leaf |
| 4 | | Commelineales | *Geogenanthus poeppigii* | Leaf |
| 5 | | | | Stem |
| 6 | | Poales | *Pharus latifolius* | Leaf |
| 7 | | | *Flagellaria guineensis* | Leaf |
| 8 | | | | Stem |
| 9 | | | *Brachypodium distachyon* | Leaf |
| 10 | | Arecales | *Euterpe oleracea* | Leaf |
| 11 | | | | Rachis |
| 12 | | Asparagales | *Semele androgyna* | Leaf |
| 13 | | | | Stem |
| 14 | | | *Geitonoplesium cymosum* | Leaf |
| 15 | | | | Stem |
| 16 | | Liliales | *Lilium primulinum* var. poilanei | Leaf |
| 17 | | | | Stem |
| 18 | | Pandanales | *Vellozia candida* | Leaf |
| 19 | | | | Stem |
| 20 | | | *Carludovica palmata* | Leaf |
| 21 | | | | Stem |
| 22 | | Dioscoreales | *Tacca integrifolia* | Leaf |
| 23 | | | | Petiole |
| 24 | | | *Tacca chantrieri* | Leaf |
| 25 | | | | Petiole and Midrib |
| 26 | | Alismatales | *Anthurium acaule* | Leaf |
| 27 | | | | Midrib |
| 28 | | | *Philodendron insigne* | Leaf |
| 29 | | | | Petiole and Midrib |
| 30 | | Acorales | *Acorus gramineus* 'Ogon' | Leaf |
| 31 | | | | Root |
| 32 | ANA-grade | Austrobaileyales | *Austrobaileya scandens* | Leaf |
| 33 | | | | Petiole and Midrib |
| 34 | | | *Kadsura borneensis* | Leaf |
| 35 | | | | Petiole and Midrib |
| 36 | | Nymphaeales | *Nymphaea thermarum* | Leaf |
| 37 | | | | Petiole |
| 38 | | Amborellales | *Amborella trichopoda* | Leaf |
| 39 | | | | Stem |

For xyloglucan digestion, PD-10 eluate was digested with an excess amount of xyloglucanase [*Aspergillus aculeatus* XEG, GH12] at 37°C overnight (1 μl enzyme was used for the 1 mg AIR equivalent PD-10 eluate).

## Polysaccharide analysis by carbohydrate gel electrophoresis (PACE)

PACE was carried out as previously described [33] with modification. In short, enzyme digestion products were labelled with 8-aminonaphthalene-1,3,6-trisulfonic acid (ANTS). After

drying samples, the derivatives were resuspended in 50 μl of 3M urea. In eudicot and ANA-grade samples, 15 μg AIR equivalents were loaded to 20% acrylamide resolving gel. In monocot samples, 45 μg AIR equivalents were loaded to the same gel. The electrophoresis was carried out at 200 V for 30 minutes and then 1000 V for two hours and 15 minutes.

## Arabidopsis mutant production

GT47A-VII expression vectors were constructed by GoldenGate-based cloning. The coding sequences of *Os*GT47A-VII, *Atr*GT47A-VII, and *At*MBGT1 were prepared by a commercial gene synthesis service (Integrated DNA Technologies, IA, USA. The nucleotide sequences are in S1 Table in S1 File). At level 1 assembly, GT47A-VII parts were assembled with 35S promoter, 3×Myc-tag, Nos terminator, and plasmid backbone. At level 2 assembly, each level 1 transcriptional unit was assembled with backbone and two other transcriptional units that encode eGFP driven by oleosin promoter and the kanamycin resistance gene driven by 35S promoter. Vectors were transformed by the floral dip method using the Agrobacterium GV3101 strain [34]. Transformants were selected by seed eGFP signal and kanamycin resistance. Genotyping was carried out using PCR (Primers are in S2 Table in S1 File).

## Rice mutant production

The plasmid for the CRISPR-Cas9 system was constructed following a previous publication [35]. Agrobacterium-mediated rice transformation (parental line: Taichu-65 variety) and antibiotic selection (hygromycin) were carried out using services provided by the Biotechnology Centre at Akita Prefectural University.

## Western blotting

Aerial parts of four week old Arabidopsis were collected. A microsome fraction was extracted from the total aerial parts following the previously described protocol [36]. 5 mg of microsome fraction was mixed with loading dye (NuPAGE™, Thermo Fisher Scientific, MA, USA) and a reducing agent (NuPAGE™ Sample Reducing Agent, Thermo Fisher Scientific, MA, USA). The mixture was heated at 95˚C for 15 minutes. The total volume of samples was applied to precast gel (Mini-Protean TGX Gels, 4–15% gel, Bio-Rad, CA, USA). SDS-PAGE was carried out following manufacturer instructions. Proteins were transferred to nitrocellulose membrane (iBlot™ Transfer Stack, Thermo Fisher Scientific, MA, USA). Ab9106 anti-Myc antibody (Abcam, Cambridge, UK) and goat anti-rabbit IgG-HRP conjugate (Bio-Rad CA, USA)) were used for the detection. Chemiluminescence was detected using Amersham™ ECL™ Detection Reagents (Cytiva, Amersham, UK) using a ChemiDoc™ (Bio-Rad CA, USA).

# Results

## Non-eudicot angiosperm galactoglucomannan composition is distinct from that found in eudicots

To examine the mannan structures oresent in non-eudicot angiosperms, we conducted polysaccharide analysis by PACE with a panel of mannan-active glycoside hydrolases. After *Cj*Man26 digestion, mannan-derived oligo saccharides appeared with variation between samples (Fig 2A). These bands were absent in no-enzyme controls (Fig 2B, S1 Fig in S1 File). In tomato fruits (as a reference eudicot species), β-GGM digestion products (GAGM, GBGM, GAGAGM, and GBGAGM) and AcGGM digestion products (GMM and MM) were observed (nomenclature is explained in Fig 1A). In all ANA-grade samples and some monocot samples (these are prepared from primary cell wall-rich tissue leaf except for *Nymphaea* as noted in

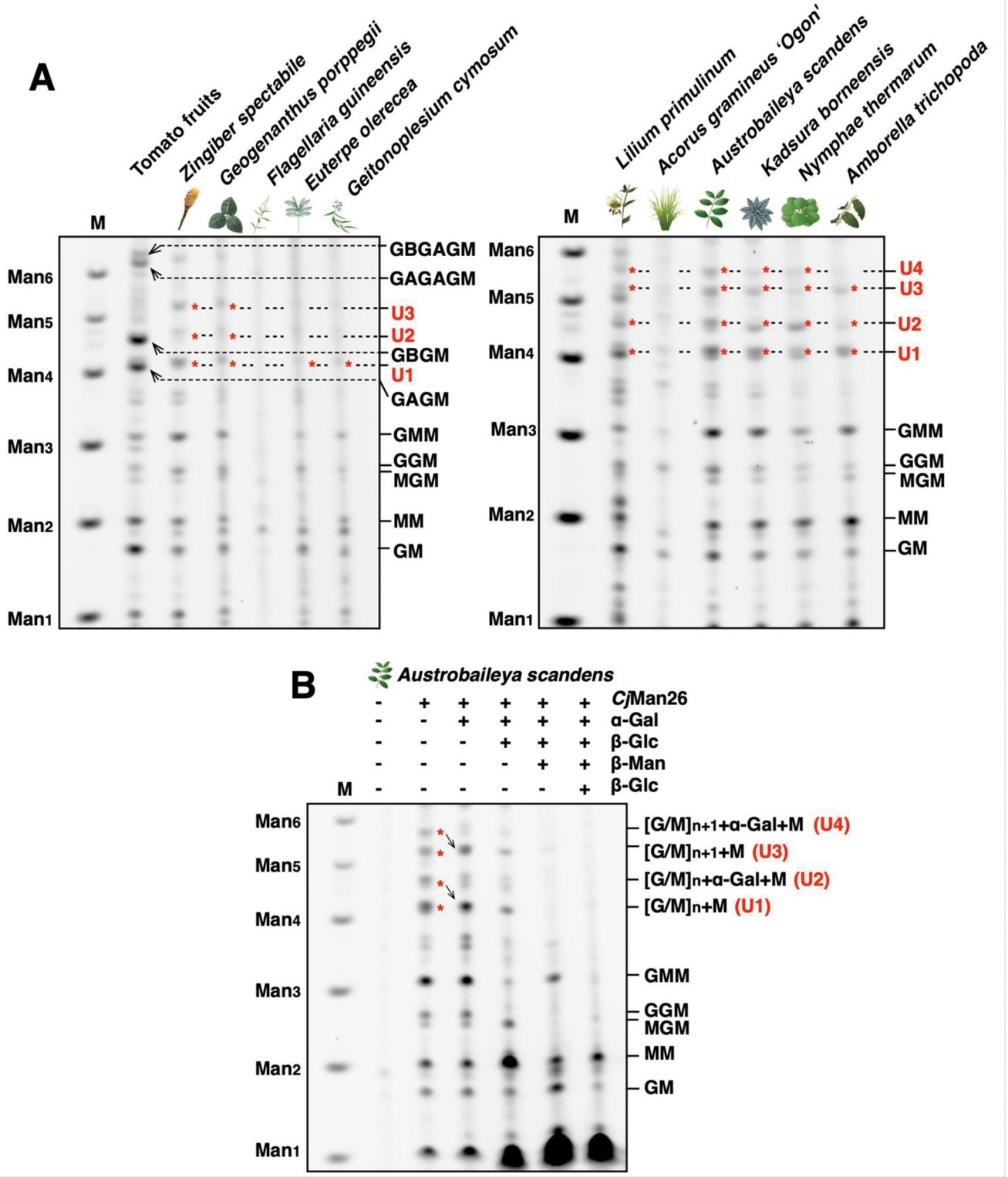

**Fig 2. Diverse mannan structures in angiosperms.** (A) Comparison of *Cj*Man26 digestion products. Tomato fruits have a digestion pattern containing both β-GGM and galactoglucomannan. GAGM, GBGM, GAGAGM, and GBGAGM arise from β-GGM. GMM, MM arise from galactoglucomannan. Some monocots and ANA-grade have unannotated oligos (U1-4, red asterisks). The samples shown here are of leaf origin except for tomato (fruits) and *Nymphae thermarum* (petiole). M: marker lane (standard oligo saccharides). (B) Sequential digestion of *Cj*Man26 digestion products from *Austrobaileya scandens* (leaf). U2 and U4 migrated to U1 and U3, respectively after α-galactosidase treatment. U1 and U3 were partially digested after β-glucosidase

treatment and completely degraded by the following β-mannosidase treatment. The number of "n" is estimated to be three, based on the migration. M: marker lane (standard oligo saccharides).

Fig 2 legend), distinctive bands appeared ranging in size between mannotetraose and manno-hexaose. We named these unknown bands as U1, U2, U3, and U4 from faster to slower migration. To characterise these structures, we carried out sequential enzyme digestion of the *Austrobaileya scandens* sample as a representative. U2 and U4 migrated to U1 and U3, respectively, after the α-galactosidase digestion (Fig 2B). Enriched U1 and U3 were partially digested by β-glucosidase treatment, yielding MM and other minor products. Probably, U1 and U3 produced by α-galactosidase treatment contained some oligosaccharides with different Glc/Man order and ratio. Among them, one had Man at the non-reducing end remaining after β-glucosidase treatment (*e.g.* MGMM and MGGMM). Then, the residual U1 and U3 were digested by β-mannosidase treatment. Therefore, U1 and U3 are deduced to be backbone oligos composed of Glc and Man with some variation of the order (represented as $[G/M]_n+M$ and $[G/M]_{n+1}+M$, The number of "n" is estimated to be three based on the position). U2 and U4 are the oligos that have α-Gal on U1 and U3 (represented as $[G/M]_n+α\text{-Gal}+M$ and $[G/M]_{n+1}+α\text{-Gal}+M$). These results indicate that these non-eudicot angiosperms have galactoglucomannan consisting of an unpatterned backbone. In addition, no evidence was found for β-GGM in non-eudicot angiosperms.

## β-(1,2)-Gal was not detected in non-eudicot angiosperms

Next, we considered the possibility that β-(1,2)-Gal is attached to a different type of mannan in non-eudicot angiosperms. To test this hypothesis, we conducted sequential digestion using *An*GH5 mannanase and β-galactosidase. As in the previous report, GBGM band migrated to GAGM in the tomato fruit sample (Fig 3A) [4]. However, a similar migration shift was not observed in monocots and ANA-grade samples (Fig 3B–3D). Therefore, we interpreted this result as β-(1,2)-Gal was not present in *An*GH5 mannanase sensitive oligosaccharides from the studied species.

## *At*MBGT1 orthologues from rice and Amborella did not show MBGT activity *in vivo*

For the production of β-GGM, three specialised enzymatic activities are required: patterned backbone synthase, an appropriate MAGT, and a MBGT (Fig 1A). Products of MAGT activity were found even in many non-eudicot angiosperms (Fig 2B). Hence, we hypothesised the limitation of β-GGM synthesis in non-eudicot angiosperms might be due to the absence of patterned backbone synthesis activity and/or the absence of MBGT activity. Indeed, candidate *At*MBGT1 orthologues are found in GT47A-VII in a wide range of angiosperms [4]. To test the activity of candidate *At*MBGT1 orthologues from monocot and ANA-grade angiosperms, rice (*Oryza sativa*) and *Amborella trichopoda* genes respectively were chosen for further analyse. Both species have only one gene in GT47A-VII, where *At*MBGT1 is found (Fig 4A). We named rice GT47A-VII (*Os*12g38450) as "*Os*GT47A-VII" and Amborella GT47A-VII (XP_006846239) as "*Atr*GT47A-VII". In addition to *Os*GT47A-VII, rice has three genes that form a grass-specific subclade near GT47A-VII, but also near to GT47A-VI that contains the xyloglucan galactosyltransferases related to MUR3. To understand the subfunctionalisation of these genes, we investigated the published rice transcriptome data [37]. *Os*GT47A-VII was expressed in several stages of inflorescence but not highly compared to xyloglucan galactosyltransferase *Os*MUR3 (Fig 4B). In contrast to *Os*10g32160, which showed weak but relatively

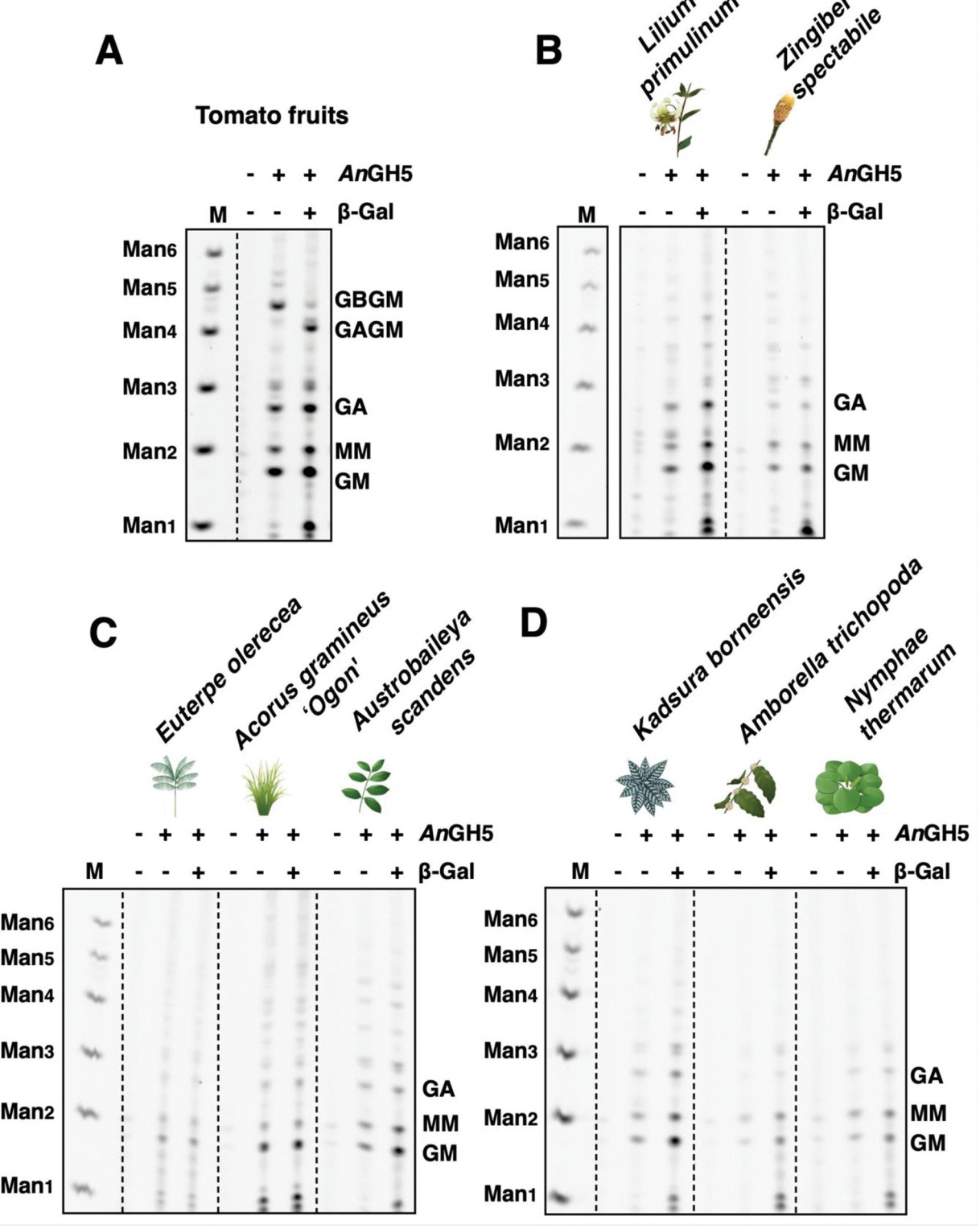

**Fig 3. β-galactosidase-specific band shifts were not observed in monocots and basal angiosperms samples.** (A) Tomato fruits produced GBGM oligo after *An*GH5 digestion, which migrated to GAGM after the following β-galactosidase treatment. M: marker lane (standard oligo saccharides). (B-D) *An*GH5 digestion products from monocots (B) and ANA-grade (C and D) showed neither GBGM nor another β-galactosidase sensitive oligosaccharide. The samples in (B-D) are of leaf origin. M: marker lane (standard oligo saccharides).

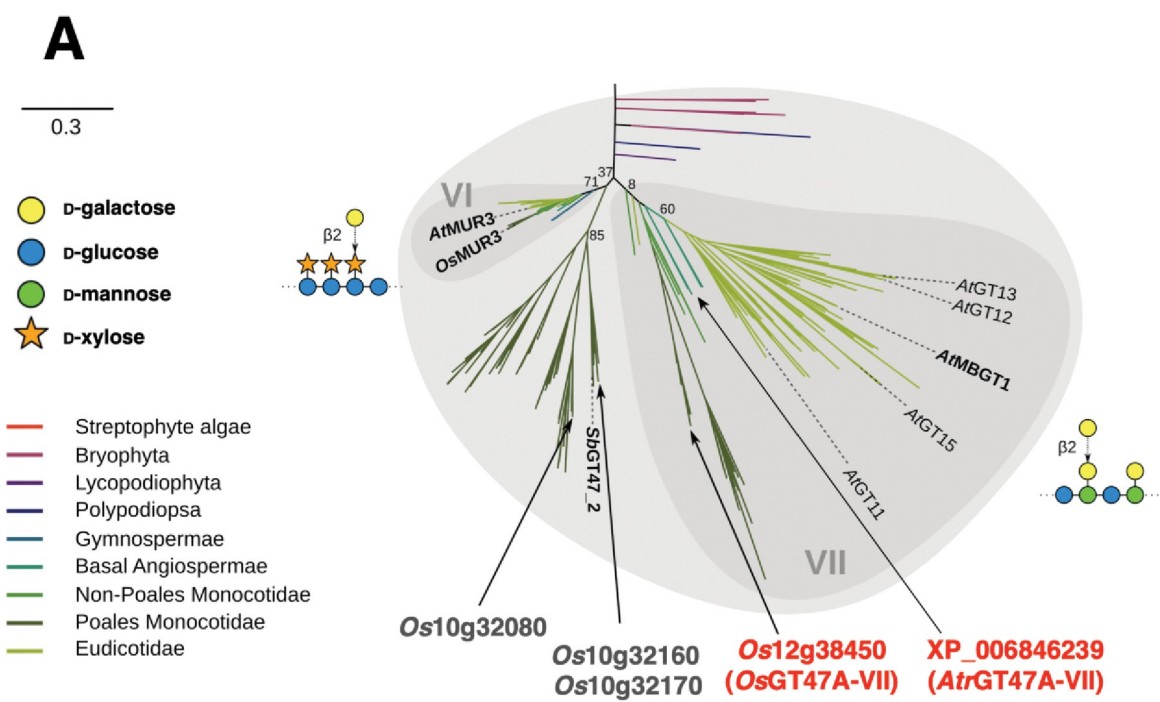

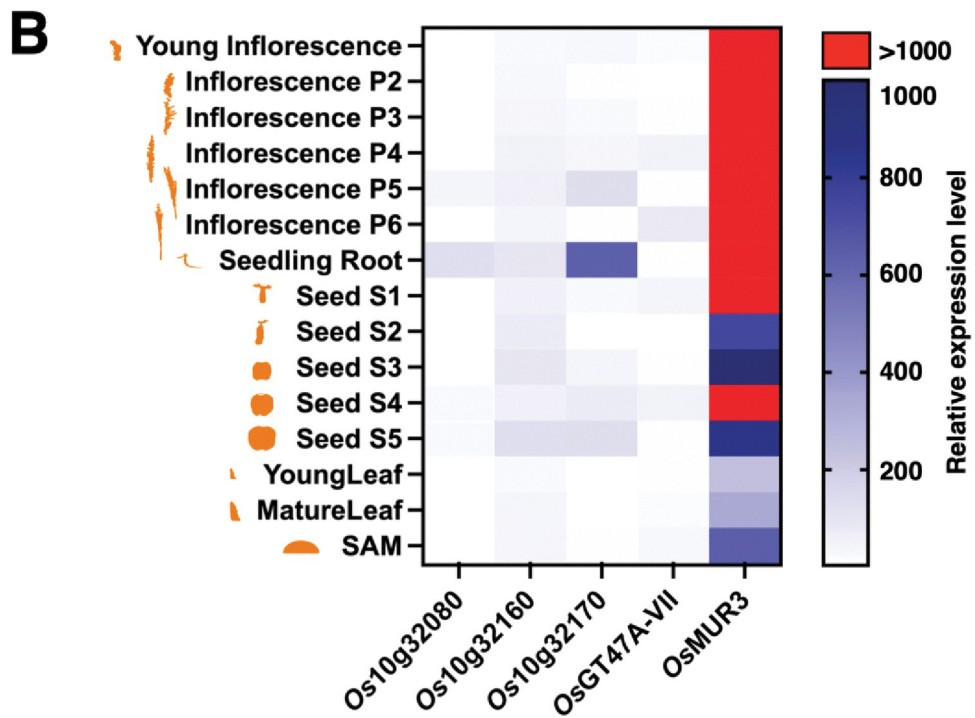

**Fig 4. *At*MBGT1 orthologues in rice and Amborella.** (A) Phylogenetic tree of GT47A-VI and VII. The tree was adapted from our previous publication [4]. *Os*GT47A-VII and *Atr*GT47A-VII were distantly located from eudicot genes in GT47A-VII subclade. (B) Expression analysis of rice genes. The expression level was presented as the median of biological triplicates. The source data were obtained in [37]. We obtained the curated form of data via the public database (bar.utoronto.ca/eplant) [46]. SAM: shoot apical meristem. Minimal data set is shown in S3 Table in S1 File.

broad expression, the closest homologue *Os*10g32170 was highly expressed in the seedling root. These divided expression patterns suggested the possibility that these GT47A genes were used for different roles spatiotemporally or had different biochemical activities.

To test the *in vivo* MBGT activity of *Os*GT47A-VII and *Atr*GT47A-VII, these two genes were expressed in the Arabidopsis *mbgt1* mutant under the control of a CaMV 35S promoter. The resulting products, fused with 3×Myc-tag, were detected by western blotting, confirming satisfactory expression (S2 Fig in S1 File). The mannan structure from young stem samples was then analysed by PACE. In the no-enzyme control, no band was observed between the mannotetraose and mannopentaose regions (S3 Fig in S1 File). However, when alkali-extract was treated with *Cj*Man26, the wild type Col-0 (hereafter referred to as WT, positive control) produced GBGM and GAGM oligosaccharides (Fig 5). On the other hand, the *mbgt1* mutant produced GAGM but not GBGM, due to lack of MBGT activity in this mutant. Introduction of *At*MBGT1 (p35S::*At*MBGT1/*mbgt1*: positive control) showed restoration of the GBGM oligosaccharide product. However, the expression of neither *Os*GT47A-VII nor *Atr*GT47A-VII allowed the production of GBGM. Therefore, we concluded that *Os*GT47A-VII and *Atr*GT47A-VII do not have MBGT activity on Arabidopsis patterned galactoglucomannan.

### *Os*GT47A-VII is a xyloglucan galactosyltransferase, which is required for proper root elongation

To evaluate further the importance and role of *Os*GT47A-VII *in vivo*, we produced two independent lines of genome-edited rice mutants (25 bp deletion in #1 and 2 bp deletion in #2 after the designed PAM site). We assessed the two-week-old plants that grew hydroponically. A distinctive growth defect was not seen in the aerial part, even though one of the two mutant lines showed a statistically significant difference in shoot length (Fig 6A). In the root, both mutants manifested shorter root (23% and 31% decrease in mean) (Fig 6B).

Since we had not obtained evidence for β-GGM or MBGT activity of *Os*47A-VII, we hypothesised that the rice root growth defect may come from a problem of xyloglucan modification. To test the activity of *Os*47A-VII on xyloglucan, we produced the complementation line using Arabidopsis *mur3 xlt2* double mutant (p35S::*Os*GT47A-VII/*mur3 xlt2*) that has almost no galactose on the xyloglucan (note: In Arabidopsis, XLT2 transfers Gal to the second Xyl from the non-reducing end of XXXG unit, while MUR3 transfers Gal to the third Xyl [38]). Xyloglucan structures of WT, *mur3*, *xlt2*, *mur3 xlt2*, and p35S::*Os*GT47A-VII/*mur3 xlt2* were analysed by PACE (Fig 6C, no enzyme control: S4 Fig in S1 File). In WT, XXXG, XXFG, and XLFG bands were dominant, and the XLXG band was faint, which is consistent with our previous publication [30]. As expected, in the *mur3* mutant the XXFG, XLFG, and XXLG bands disappeared, but XLXG appeared. In the *xlt2* mutant, the XLFG and XLXG bands disappeared. The double mutant only had XXXG. Interestingly, in the complementation line, XXXG and XXFG were seen. Further, a faint XXLG band was observed. This band pattern was similar to *xlt2* mutant. Therefore, we concluded that *Os*GT47A-VII showed xyloglucan β-galactosyltransferase activity, which is likely to have similar substrate specificity to MUR3 rather than XLT2.

## Discussion

### Distribution of β-GGM

In this study, we analysed the mannan structure of 21 species (39 samples) from monocots and the ANA-grade. Most of the cell walls contained some digestible glucomannan, producing mannan-derived oligosaccharides after mannanase digestion. Interestingly, analysis of many

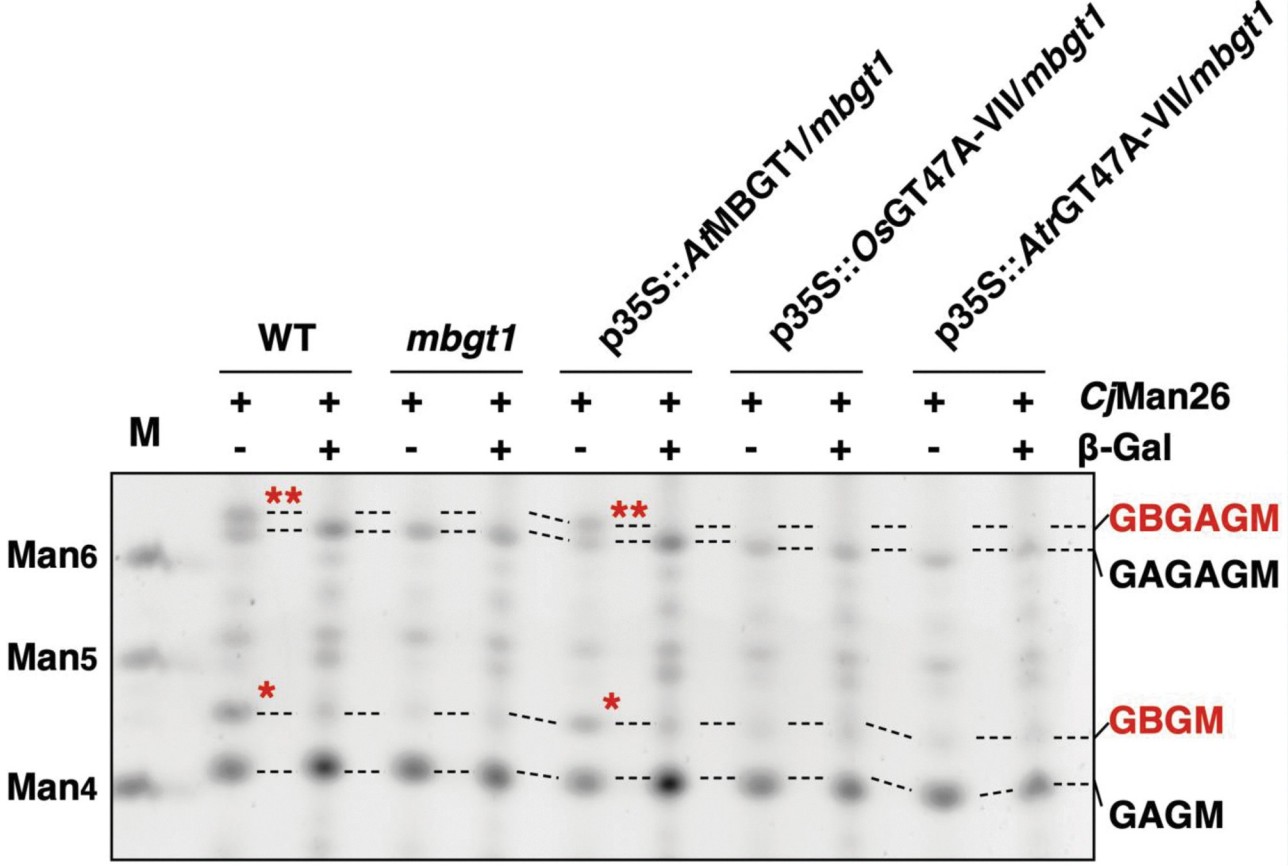

**Fig 5. Complementation analysis of rice and Amborella GT47A-VII in Arabidopsis *mbgt1*.** Analysis of *Cj*Man26 digestion products. GBGM is diagnostic for MBGT activity. While WT and p35S::*At*MBGT1/*mbgt1* have the GBGM band, *mbgt1* and two GT47A-VII introduced lines (p35S::*Os*GT4We 7A-VII/*mbgt1* and p35S::*Atr*GT47A-VII/*mbgt1*) did not produce the band. Three independent overexpression lines have been examined for the mannan structure analysis and one representative image is shown. M: marker lane (standard oligo saccharides).

of these plants generated unique oligosaccharides that were not seen in studies of our model eudicots Arabidopsis and tomato. Sequential digestion revealed that the backbone of unique products is composed of a distinct unpatterned sequence of Glc and Man, which may arise from AcGGM synthesised by CSLA enzymes with different activity to that in the eudicots. In addition, the MAGTs may have different substrate specificity between Arabidopsis and non-eudicot angiosperms. Many monocot and ANA-grade cell walls contained glucomannan that is α-galactosylated (Fig 2B), in contrast to Arabidopsis AcGGM, which is not substantially α-galactosylated [4]. Taken together, eudicots and non-eudicot angiosperms have different backbone structures and α-galactosylation frequency of AcGGM.

To expand the knowledge of β-GGM distribution in plant lineages, we investigated the presence of β-GGM in sister groups of eudicots. Interestingly, oligosaccharides with a β-Gal side chain were not observed. There were some unavoidable technical limitations in this study. Firstly, we could analyse a limited number of species of monocots and basal angiosperms. Secondly, β-GGM might exist in unsampled tissues, as we were mainly studying leaves where β-GGM is found in eudicots. Thirdly, unexpected side chain structure could hinder the digestion by *Cj*Man26 or *An*GH5, which makes oligosaccharides invisible by PACE analysis. Future structural studies should analyse additional species and tissues. Nevertheless, these results suggest that β-GGM is a eudicot-specific mannan.

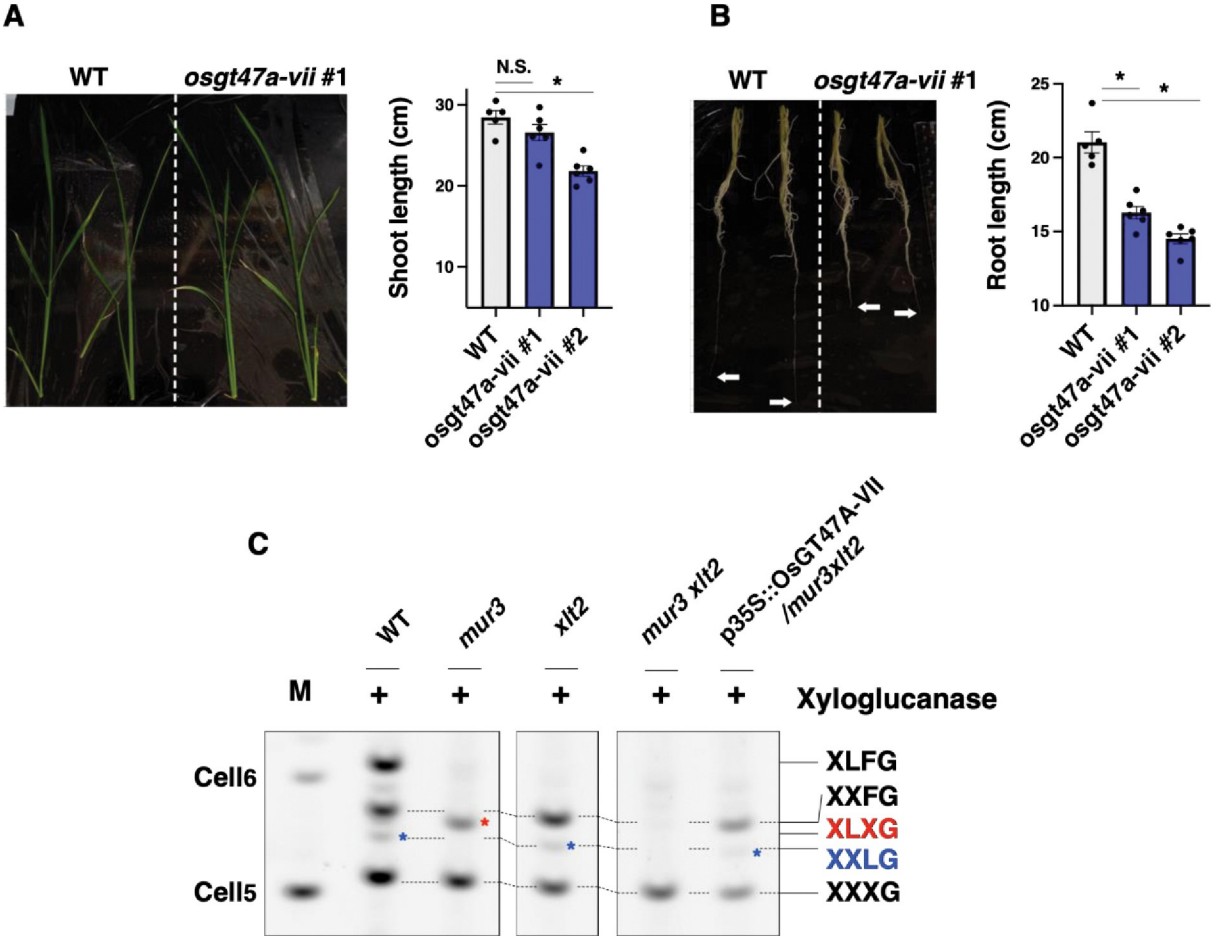

**Fig 6. *Os*GT47A-VII is a galactosyltransferase on xyloglucan.** (A) Shoot growth phenotype in rice mutant (*osgt47a-vii*). Statistical test: t-test (n = 5~6. N.S. not significant. *P<0.05). (B) Root growth phenotype in rice mutant (*osgt47a-vii*). Statistical test: t-test (n = 5~6. *P<0.05). Minimal data set is shown in S4 Table in S1 File. (C) Xyloglucan structure analysis after XG5 xyloglucanase digestion (red asterisk: XLXG band, blue asterisk: XXLG band). Three independent overexpression lines have been examined and one representative image is shown. M: marker lane (standard oligo saccharides).

## MBGT activity in GT47A-VII

If β-GGM is synthesised outside eudicot lineages, MBGT enzymes may expect to be identifiable. However, the closest rice and Amborella GT47A-VII orthologues to the single known MBGT from Arabidopsis did not show MBGT activity in complementation experiments. The complementation experiment in this study showed that MBGT activity is not common to all enzymes in GT47A-VII. In other words, *At*MBGT1 activity arose in a lineage-independent manner. This idea is supported by the activity of *At*GT11. *At*GT11 is one of the *At*MBGT1 homologues but it is implicated as pollen tube-specific xyloglucan galactosyltransferase [39]. Similarly, the present study confirmed the activity of *Os*GT47A-VII to xyloglucan (Fig 6C). Given that the GT47A-VII shares the common ancestor with the GT47A-VI (xyloglucan galactosyltransferase MUR3 subclade), it is likely that the original activity of GT47A-VII activity is xyloglucan galactosyltransferase. Gene duplication and diversification of GT47A-VII in Arabidopsis and other eudicots shown in the phylogenetic tree (Fig 4A), may have lead to the emergence of an enzyme that accommodates patterned galactoglucomannan as the acceptor.

In terms of the functional role of *Os*GT47A-VII, the stunted root phenotype of the CRISPR mutant was interesting because the amount of xyloglucan in grass species is significantly lower compared to eudicots [40]. However, differences in xyloglucan structure by tissue in rice have been reported. In rice shoot, XXGG or XXGGG type structures were found but not XXXG, while in root, XXXG type structure (fucosylated xyloglucan) was detected [41]. This characteristic difference matches the mutant growth phenotype (Fig 6A). These results indicate that the side-chain modification of xyloglucan in specific tissue is functionally important in grass species.

It is known that the donor preference of GT47A does not always correspond to the subclade. For example, both xyloglucan galactosyltransferase and xyloglucan arabinopyranosyltransferase sit in the GT47A-I clade [42]. Similarly, xyloglucan galactosyltransferase XLT2 activity was found in at least three subclades in GT47A [43–45]. Therefore, we cannot exclude the possibility that MBGT activity also arose in different subclades of GT47A. If functional differentiation between β-GGM and xyloglucan is biologically important in eudicots, plants of a different lineage, which presumably do not have β-GGM, may be more functionally dependent on xyloglucan or other polysaccharide such as mixed-linkage glucan in the grass family.

## Supporting information

**S1 File.**
(PPTX)

## Acknowledgments

We would like to thank the University of Cambridge Botanic Garden for providing monocots and ANA-grade samples. We also thank Ryo Yoshida for drawing beautiful plant illustrations and waiving their copyright.

## Author Contributions

**Conceptualization:** Konan Ishida, Yoshihisa Yoshimi, Louis F. L. Wilson, Li Yu, Paul Dupree.

**Funding acquisition:** Hiroaki Iwai, Paul Dupree.

**Investigation:** Konan Ishida, Yusuke Ohba, Alberto Echevarría-Poza, Hiroaki Iwai, Paul Dupree.

**Methodology:** Louis F. L. Wilson, Paul Dupree.

**Supervision:** Yoshihisa Yoshimi, Li Yu, Hiroaki Iwai, Paul Dupree.

**Writing – original draft:** Konan Ishida.

**Writing – review & editing:** Yoshihisa Yoshimi, Louis F. L. Wilson, Li Yu, Hiroaki Iwai, Paul Dupree.

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
