## [Decision Letter · Decision Letter 0]

13 Aug 2023

PONE-D-23-22384Differing structures of galactoglucomannan in eudicots and non-eudicot angiospermsPLOS ONE

Dear Dr. Dupree,

Thank you for submitting your manuscript to PLOS ONE. After careful consideration, we feel that it has merit but does not fully meet PLOS ONE’s publication criteria as it currently stands. Therefore, we invite you to submit a revised version of the manuscript that addresses the points raised during the review process. One of the reviewers has also offered his suggestions to improve the text (see his attached file).

We look forward to receiving your revised manuscript.

Kind regards,

Olga A. Zabotina, PhD

Academic Editor

PLOS ONE

Journal Requirements:

3. Please expand the acronym “UKRI” (as indicated in your financial disclosure) so that it states the name of your funders in full.

6. We notice that your supplementary [figures/tables] are included in the manuscript file. Please remove them and upload them with the file type 'Supporting Information'. Please ensure that each Supporting Information file has a legend listed in the manuscript after the references list.

Reviewers' comments:

Reviewer's Responses to Questions

**Comments to the Author**

1. Is the manuscript technically sound, and do the data support the conclusions?

Reviewer #1: Yes

Reviewer #2: Yes

2. Has the statistical analysis been performed appropriately and rigorously? 

Reviewer #1: Yes

Reviewer #2: I Don't Know

3. Have the authors made all data underlying the findings in their manuscript fully available?

Reviewer #1: Yes

Reviewer #2: Yes

4. Is the manuscript presented in an intelligible fashion and written in standard English?

Reviewer #1: Yes

Reviewer #2: Yes

5. Review Comments to the Author

Reviewer #1: This rigorous study builds on the Dupree lab's recent and exciting discovery about the structure of patterned beta-galactosylated glucomannans (abbreviated β-GGM) in multiple eudicot species, including Arabidopsis and tomato. Here, the authors investigated the occurrence of β-GGM in 17 monocot species and four basal angiosperm species. Using a series of enzymatic digestions for polysaccharide analysis by gel electrophoresis, the non-eudicot plant tissues did not show any of the carbohydrate bands diagnostic of β-Gal-decorated mannan structures in tomato fruits. While α-Gal mannan substitutions were present, sequential enzyme treatments did not reveal any conserved glycosyl residue patterns in Austrobaileya scandens glucomannan. β-Galactosidase treatments did not alter the GH5 mannanase-released oligosaccharide patterns for the non-eudicot species. Consistent with this observation, putative orthologs of the AtMBGT1 β-galactosyltransferase from Amborella trichopoda and rice (Oryza sativa) did not complement the chemotype of the Arabidopsis mbgt1 mutant. Instead, mutations in the rice gene OsGT47A-VII and its overexpression in Arabidopsis mur3 xlt2 double mutant showed phenotypes consistent with MUR3-like β-Gal addition to the xyloglucan backbone.

Altogether, this study indicates that MAGT1 substrate specificity and β-GGM presence are likely eudicot-specific. This manuscript is excellent, so I only have a couple of minor suggestions:

line 284 – “manifested growth defects” – the text does not indicate what was decreased

lines 314-315 “AcGGM (CSLA9 mannan)” this brief phrase should be re-worded or expanded to assist readers since AcGGM biosynthesis would require multiple activities beyond CSLA9

Fig. 4A the legend includes sugars that are not displayed elsewhere in the figure

Reviewer #2: This manuscript presents an investigation of mannan structures among diverse plant species conducted using PACE with various glycanases and glycosidases. In previous work, the authors identified beta-GGM in Arabidopsis, and in this study they aimed to determine the phylogenetic distribution in plants.

In this study, PACE analysis revealed no evidence of beta-GGM in the tissues of species tested. The authors acknowledge that while they found no evidence of beta-GGM among the tested plants that this structure might still be present in non-tested tissues, etc.

Complementary to PACE analyses, the authors also investigated the functions of homologs of AtMBGT1. Rice and Amborella MBGT-like sequences did not complement the atmbgt1 mutant. Rice mutants containing lesions in the OsGT47A-VII gene exhibited defects consistent with deficits in xyloglucan galactosyltransferase activity.

Overall, the experiments seem to have been conducted well and the results are fairly clear. The reviewer's most consequential concern is that some of the patterns on the PACE gels were very faint, making them difficult to interpret (e.g., Fig 3 C & D). What if, for example, the quantity of beta-GGM is low in the sample, might analyzing more sample provide a more convincing result? Could the authors show more heavily loaded PACE gels? It could also be helpful to have an estimate of how small a quantity of beta-GGM is needed for detection (i.e., detection limit). In this way, the authors could further qualify their inability to detect this carbohydrate and present an upper limit for the amount that might be present.

Aside from this, I have provided a marked up version of the manuscript with many comments where revisions might improve the clarity of presentation, flow and conventional use of the language. There were a number of spots in the Materials and Methods section where more details would be needed to replicate the experiments - this should be addressed. I hope my writing is legible and the feedback helpful for implementing revisions.

6. PLOS authors have the option to publish the peer review history of their article (what does this mean?). If published, this will include your full peer review and any attached files.

Reviewer #1: **Yes: **Catalin Voiniciuc

Reviewer #2: **Yes: **Aaron Liepman

---

## [Author Response · Author response to Decision Letter 0]

28 Sep 2023

manuscript: PONE-D-23-22384

Dear Dr Zabotina,

Thank you for handling our revised manuscript “Differing structures of galactoglucomannan in eudicots and non-eudicot angiosperms” for publication in PLOS ONE. We appreciate the time and effort that you and the reviewers dedicated to providing feedback on our manuscript. We have incorporated the suggestions made by the reviewers as much as possible. 

Yours sincerely,

Paul Dupree

Detailed Response to reviewers

Reviewer #1: This manuscript is excellent, so I only have a couple of minor suggestions:

line 284 – “manifested growth defects” – the text does not indicate what was decreased.

We appreciate your high regard for the manuscript. 

We corrected the sentence as below.

“In the root, both mutants manifested shorter seminal roots (23% and 31% decrease in mean) (Fig 6B).”

lines 314-315 “AcGGM (CSLA9 mannan)” this brief phrase should be re-worded or expanded to assist readers since AcGGM biosynthesis would require multiple activities beyond CSLA9.

We removed “(CSLA9 mannan)” from the sentence.

Fig. 4A the legend includes sugars that are not displayed elsewhere in the figure.

We removed the sugars that are not used in the figure from the legend.

Reviewer #2: Overall, the experiments seem to have been conducted well and the results are fairly clear. The reviewer's most consequential concern is that some of the patterns on the PACE gels were very faint, making them difficult to interpret (e.g., Fig 3 C & D). What if, for example, the quantity of beta-GGM is low in the sample, might analyzing more sample provide a more convincing result? Could the authors show more heavily loaded PACE gels? It could also be helpful to have an estimate of how small a quantity of beta-GGM is needed for detection (i.e., detection limit). In this way, the authors could further qualify their inability to detect this carbohydrate and present an upper limit for the amount that might be present.

We appreciate your detailed reading of the manuscript and feedback.

We understand the concern about the intensity of the bands. All forms of glucomannan are relatively low abundance in monocots compared with other plants, meaning that the signals were often scarcely above background. Loading more sample is not necessarily a good idea as it increases the background intensity. The strength of the observation of absence of bands lies in the large number of samples- replicates and different species and tissues analysed, many of which are shown here to illustrate the observation. Had any sample shown clear beta-GGM bands as we find in all eudicots we studied, then this would be apparent we believe. Since beta-GGM is only a few percent of cell walls (Yu et al. 2022) this means our detection method is sensitive.

Regarding Fig 3, the experimental setting is designed to discriminate the qualitative phenomenon of the presence or absence of a shift from the band containing the B side chain to the band containing the A side chain. The band intensities are not high (saturated) to make the increase in band intensity clear after galactosidase treatment.

Aside from this, I have provided a marked up version of the manuscript with many comments where revisions might improve the clarity of presentation, flow and conventional use of the language. There were a number of spots in the Materials and Methods section where more details would be needed to replicate the experiments - this should be addressed. I hope my writing is legible and the feedback helpful for implementing revisions.

Thank you for these helpful suggestions. According to your suggestions, we corrected our manuscript as much as possible. However, some points were not changed. The reasons for these are explained below.

・About the enzyme unit

We couldn’t write specific enzyme unit because some of the enzymes are made on a non-commercial collaboration basis and we did not calculate an enzyme unit. We optimised with each enzyme the reaction conditions (amount of enzyme, reaction temperature, and reaction time) for the complete digestion of glucomannan- so the enzyme is in great excess. 

・Regarding the schematic diagrams in Fig 1 and Fig 6

For Fig 1, a legend for the constituent sugars has been added. For the structure of xyloglucan in Fig. 6, we did not add a model diagram of the structure, as we believe that the nomenclature is well established in the field.

---

## [Editor Report · Decision Letter 1]

30 Oct 2023

Differing structures of galactoglucomannan in eudicots and non-eudicot angiosperms

PONE-D-23-22384R1

Dear Dr. Dupree,

We’re pleased to inform you that your manuscript has been judged scientifically suitable for publication and will be formally accepted for publication once it meets all outstanding technical requirements.

Kind regards,

Olga A. Zabotina, PhD

Academic Editor

PLOS ONE
---

## [Editor Report · Acceptance letter]

7 Nov 2023

PONE-D-23-22384R1 

Differing structures of galactoglucomannan in eudicots and non-eudicot angiosperms 

Dear Dr. Dupree:

I'm pleased to inform you that your manuscript has been deemed suitable for publication in PLOS ONE. Congratulations! Your manuscript is now with our production department. 

Kind regards, 

on behalf of

Dr. Olga A. Zabotina 

Academic Editor

PLOS ONE